# “It Would Ruin My Life”: Pacific Islander Male Adolescents’ Perceptions of Mental Health Help-Seeking—An Interpretative Phenomenological Analysis Focus Group Study

**DOI:** 10.3390/ijerph22010062

**Published:** 2025-01-04

**Authors:** Melia Fonoimoana Garrett, Elizabeth A. Cutrer-Párraga, G. E. Kawika Allen, Ellie L. Young, Kristofer J. Urbina, Isabel Medina Hull

**Affiliations:** 1School Psychology Department, Nebo School District, Spanish Fork, UT 84660, USA; melia.fonoimoana@gmail.com; 2Counseling Psychology & Special Education, Brigham Young University, Provo, UT 84602, USA; gekawika_allen@byu.edu (G.E.K.A.); ellie_young@byu.edu (E.L.Y.);; 3Special Education Department, Alpine School District, American Fork, UT 84003, USA; isabelmhull@gmail.com

**Keywords:** Pacific Islanders, Polynesians, mental health, mental wellness, interpretative phenomenological analysis

## Abstract

Given rising mental health concerns among Pacific Islander (PI) adolescents, this population remains underserved by available mental health resources. This interpretative phenomenological analysis with focus groups (IPA-FG) examined the lived experiences and perceptions of 19 male PI adolescents (ages 14–16) from Native Hawaiian, Maori, Samoan, and Tongan backgrounds regarding mental health help-seeking behaviors. Four overarching themes emerged: stigma and judgment, cultural misalignment in professional services, impact of disclosure and perceived punitive consequences, and a complex ecosystem of trusted relationships as mental health support. Notably, participants expressed belief that disclosing mental health challenges would “ruin their lives” and held misconceptions about adults’ ability to address mental health concerns. These findings are particularly significant given high suicidality rates among PI adolescents. This study provides insights for developing culturally responsive mental health interventions and highlights the urgent need to address mental health stigma within PI communities. Implications for practice are discussed.

## 1. Introduction

In 2016, 11.8 million US adults reported unmet mental health needs, citing barriers like cost, lack of time, uncertainty about services, or the belief that issues would resolve on their own [1]. Avoiding mental health care often leads to worsening conditions [2]. Research indicates that untreated mental health issues in children, such as anxiety, depression, self-harm, eating disorders, and hyperkinetic disorders, tend to persist into adulthood, often resulting in long-term challenges [3,4]. Without intervention, conditions like depression and anxiety can significantly affect both mental and physical health [5,6].

Among minoritized groups in the United States (US), such as Asian American, African American, and Hispanic communities, utilization rates for mental health services are significantly lower than those of White communities [7,8]. While there is research on barriers to mental health care among these groups [9], Pacific Islanders (PIs) are notably underrepresented in these studies. Most research groups PIs with Asian Americans, despite significant cultural and physiological differences between the two populations [10,11,12]. This lack of specific research highlights the need to better understand the barriers PIs face in accessing mental health services.

PIs refer to individuals from regions such as Hawaii, Samoa, Guam, Fiji, Tonga, and the Marshall Islands, as well as territories in Melanesia, Micronesia, and Polynesia under US jurisdiction [13]. In this study, the term “PI” encompasses related terms like Native Hawaiian Pacific Islander (NHPI) and Polynesian American [14], as they represent the same ethnic groups.

### 1.1. Statement of the Problem

Although mental health disorders are widespread in the US, minoritized individuals access mental health services at much lower rates than White individuals [15,16]. This is concerning, as minoritized populations are often exposed to additional risk factors—such as lower socioeconomic status, discrimination, and unequal access to education—that increase their vulnerability to mental health challenges [17,18,19,20,21].

The PI community faces heightened risks for depression, suicide, and stigma surrounding mental health care, though research in these areas remains limited [22,23]. PI high school students may be especially vulnerable due to the combination of adolescent risk factors and barriers unique to their community [24,25,26,27,28]. However, there is insufficient data on their perceptions, attitudes, and barriers to mental health care. Researchers note that despite the growth of Asian American and Native Hawaiian and PI populations, data on these groups often lack integrity due to small sample sizes, leading to their exclusion from critical research [29].

### 1.2. Importance of Mental Health Services

Mental health issues are widespread in the US, with 20.6% of adults (51.5 million) reporting a mental illness. Among them, only 44.8% received mental health services in the prior year [15]. Effective treatments, including therapy and medication, can significantly improve the quality of life for individuals suffering from anxiety and depression and reduce the incidence of self-harm and suicidal behavior [30,31,32,33,34].

Left untreated, mental health disorders can become chronic and lead to severe psychological, social, and physical consequences. Depression is often recurrent [35] and is linked to various physical ailments, such as coronary heart disease and diabetes [5,6]. Expanding access to mental health services is essential for maintaining mental and physical well-being.

Despite this, many needs remain unmet. In 2016, nearly 12 million Americans reported requiring but not receiving mental health services [1]. Globally, underutilization is common, as seen in studies across Australia and Europe [36,37,38]. Increased access to mental health care could alleviate these disparities, as psychotherapy is proven effective for various conditions, including anxiety, depression, and panic disorder [39].

### 1.3. Minorities and Mental Health Care

Minoritized populations experience mental health disorders at similar rates to the majority population [40,41] but face additional risk factors, including socioeconomic disparities, perceived discrimination, and lower-quality education [17,19,21]. Asian Americans and Pacific Islanders (AAPIs) encounter unique challenges, such as acculturative stress, stigma, and shame, which further hinder mental health service utilization [16,23].

Barriers to accessing care include stigma, distrust of providers, lack of insurance, and experiences of racism within health care systems [42]. Despite these challenges, research demonstrates the effectiveness of therapy across racial and ethnic groups, including African Americans, AAPIs, Latinos/as, and Native Americans [43]. Additional research is needed to understand and address the barriers faced by minoritized populations.

### 1.4. Effects of Underutilization

Underutilization of mental health services results in delayed intervention, which often leads to poorer long-term outcomes [44,45]. Unfortunately, disparities persist across care settings. For instance, minoritized groups often experience higher rates of involuntary inpatient care but lower rates of outpatient treatment [46,47]. PIs, in particular, exhibit high rates of mental health needs but avoid or delay services due to cultural factors [11], and 36% report significant mental health concerns and 26% avoiding treatment [48].

#### 1.4.1. Stigma and Perceptions of Mental Health Services

Stigma remains a major barrier to mental health care for minoritized populations. High levels of perceived public stigma are strongly associated with a reduced likelihood of seeking help [16,49]. This stigma often manifests through community attitudes, stereotypes, and discrimination, which exacerbate the underutilization of mental health services [50]. Notably, the perception of stigma within these communities may be higher than the actual level of stigma present, yet its negative impact on help-seeking behavior remains profound [49].

In addition to stigma, negative perceptions of mental health services further deter utilization. African Americans, for instance, may believe mental health issues resolve without intervention, contributing to lower service usage compared to White counterparts [51]. Furthermore, minoritized populations frequently report heightened levels of fear and distrust toward mental health professionals, rooted in both historical and contemporary experiences of systemic discrimination [52,53]. Together, these factors contribute to persistent disparities in mental health service access and outcomes.

#### 1.4.2. Availability of Mental Health Services

Minoritized groups often lack access to high-quality mental health care. Research indicates these populations receive services less frequently and of lower quality compared to the White majority [54]. Language barriers further complicate access, as only 10.8% of practitioners report offering services in languages other than English [55]. Addressing these systemic inequities is critical to reducing disparities and improving care for all.

#### 1.4.3. Cultural Fit Within Therapy

Research has shown that ethnic matching in therapy can increase treatment utilization and reduce dropout rates [56]. Clients paired with practitioners from their minoritized communities are less likely to require crisis intervention, emergency services, or inpatient care [57]. However, a significant disparity exists, as 86% of mental health practitioners identify as White, despite only 60% of the US population being White [58,59]. This limits options for minoritized individuals seeking culturally congruent therapy.

Individuals from minoritized populations often prefer practitioners familiar with their culture, resulting in higher rates of service continuation and improved outcomes [60]. For PIs, clinicians with cultural competence yield better results, particularly when addressing presenting concerns and therapy outcomes [10,14,22].

### 1.5. Student Mental Health Challenges

High school students increasingly face mental health challenges, with 70% of adolescents identifying anxiety and depression as major concerns among their peers [61]. These challenges stem from academic pressures, familial expectations, bullying, and substance use [61,62]. The scope of this issue is significant: over 27% of individuals aged 3–17 have been diagnosed with Attention Deficit Hyperactivity Disorder (ADHD), anxiety, or depression [63], yet approximately 60% of adolescents with major depression receive no treatment [64]. Mental health challenges frequently lead to school disengagement and dropout, particularly when compounded by severe bullying, social isolation, or family problems [65,66].

For minoritized students, these challenges are intensified by additional barriers including stigma, insufficient resources, and systemic inequities. Even when mental health care is accessible, White students are more likely than their minoritized peers to receive depression diagnoses and treatment [67]. Discrimination, bullying, and lower socioeconomic status contribute to higher rates of depression and unmet mental health needs among minoritized youth [24,28]. Most concerning, instead of receiving adequate mental health support, these students often face punitive measures such as suspension or incarceration [68]. Studies show 65–70% of youth in juvenile detention are from minoritized groups, with many receiving their first mental health services while incarcerated [69,70].

### 1.6. PIs Within the United States

The US is becoming increasingly diverse, with over 40% of the population identified as part of a minoritized group [59]. While the White population remains the majority, it has decreased in number, whereas other ethnic groups, particularly Native Hawaiians and PIs, have seen significant growth. The PI population grew by 61% between 2010 and 2020, marking the third-fastest increase among ethnic groups.

Despite this growth, PIs exhibit the lowest rates of help-seeking behavior for mental health care. The National Survey of Drug Use and Health [71] found that 73.1% of PI adults with mental health needs did not receive treatment, compared to 56.7% of the general population [72]. Among PIs, barriers to care are exacerbated by lack of insurance; although the uninsured rate for the White population is 7.8%, the PI community experiences an uninsured rate of 9.3% [73].

Mental health risks within the PI community, including depression and suicide, are high, due to factors such as acculturative stress, generational status, and poverty [23]. These risks are compounded by racial discrimination, which significantly impacts both the mental and physical health of PIs. For example, perceived racism is linked to higher symptoms of depression among Native Hawaiians [74] and to lower self-esteem and life satisfaction and higher rates of anger, anxiety, and stress [75].

Stigma surrounding mental illness is also a significant barrier. PIs report higher levels of both self-stigma (among men) and public stigma (among women), contributing to the underutilization of mental health services [11]. This issue is especially pressing, as suicide rates have been rising within the PI community, increasing by 16% between 2014 and 2019, despite a decrease in suicide rates among the White population [76].

Although PIs face numerous challenges, they also employ various coping mechanisms to buffer psychological distress. Many rely on collectivistic coping strategies, such as seeking support from family and faith, which promote positive psychological well-being [10,14]. However, certain strategies, such as venting and behavioral disengagement, have been found to exacerbate psychological distress, particularly among Native Hawaiians [77].

Mental health services that are tailored to the cultural needs of PIs yield better outcomes for the community [14]. Professionals working with PIs can improve their effectiveness by understanding and addressing the unique cultural needs of this population [11].

#### PI Students

Male PI adolescents often define their identity through strength, power, and respect. In many cases, this manifests in violent expressions of masculinity, partly as a response to the historical oppression of their communities [78]. PI students face significant challenges in education, with the second-highest high school dropout rate among minoritized groups, at 8% [79]. Mental health concerns are often linked to school dropout, with one study finding that students with distress disorders are twice as likely to leave school early [80,81,82]. This is concerning, as high school graduation is a critical factor in securing stable employment and a livable wage [83].

Suicide rates are particularly high among PI youth, especially those aged 15–24. According to the US Department of Health and Human Services, Office of Minority Health (2019), suicide is the leading cause of death in this age group within the PI community [84]. PI youth are three times more likely to attempt suicide than their European counterparts, with adolescent PIs showing the highest suicide rates across PI ethnic age groups [85].

Bullying is another significant issue for PI students. A White House task force identified harassment as a common experience; bullying, including physical and emotional abuse, is linked to higher rates of suicidal ideation among both victims and perpetrators [26]. To address the mental health needs of PI students, it is critical to examine the risk factors and barriers to care.

PIs are underrepresented in mental health research, which often combines them with Asian Americans, despite the cultural and geographical differences between the two groups. It was not until 1997 that PI and Asian Americans were separated as distinct categories on the US Census [86]. While both groups show similar initial utilization of mental health services, PIs are less likely to remain in therapy for extended periods, and their concerns tend to focus more on family-related issues [87]. This gap in research has resulted in a lack of culturally relevant services for the PI community [10,88,89].

The underutilization of mental health services among PIs is attributed to multiple factors, including stigma, cultural misfit, and general distrust of health care systems [42,56,90]. To better meet the needs of this growing population, mental health professionals must develop a deeper understanding of the cultural factors that affect PI communities and tailor their services accordingly. Given the documented mental health challenges facing PI youth and the gaps in research specific to this population, this study aimed to understand how male PI adolescents perceive and make meaning of mental health challenges, help-seeking behaviors, and barriers to accessing mental health services. While traditional research often begins with specific hypotheses, our use of Interpretative Phenomenological Analysis (IPA) methodology [89] deliberately avoids predetermined hypotheses to allow participants’ lived experiences to emerge organically. This approach was particularly valuable given the paucity of PI-specific mental health research and the need to understand these issues through the cultural lens of PI male adolescents themselves.

## 2. Materials and Methods

### 2.1. Theoretical Orientation

This study utilized a culturally grounded PI framework, ‘Talanoa’, as its theoretical orientation [91]. Talanoa reflects traditional PI practices of knowledge-sharing and relational engagement, often translated as “talking about nothing in particular”, and signifies an open dialogue where participants share stories, experiences, and emotions within a context of mutual respect, reciprocity, and trust [92]. This theoretical foundation reflects how deeply PI values are rooted in human connection and the collective construction of meaning. In this study, Talanoa served as the lens for exploring how culture, relationships, and perceptions of mental health are interwoven within the experiences of male PI adolescents [93]. The emphasis on storytelling and co-construction of knowledge inherent in Talanoa allowed for culturally resonant engagement, and helped foster a safe space where participants could discuss sensitive topics, such as mental health, without fear of judgment or misunderstanding [93]. By framing the research within Talanoa, the study privileged PI epistemologies, ensuring that the insights generated were rooted in and reflective of participants’ cultural contexts [94].

### 2.2. Research Design

#### 2.2.1. IPA

Interpretative Phenomenological Analysis (IPA) was selected as the research design for this qualitative study examining how male PI adolescents conceptualize mental health supports. As a qualitative methodology, IPA enables researchers to deeply explore participants’ lived experiences and meaning-making processes through semi-structured interviews and rigorous analytical procedures. IPA’s origins in health care research [95,96] make it particularly suitable for investigating mental health help-seeking behaviors and experiences. Additionally, IPA has demonstrated effectiveness with PI populations in similar mental health research contexts [89].

The philosophical foundations of IPA align well with this study’s objectives through three key elements. First, its phenomenological focus enables exploration of participants’ subjective lived experiences. Second, its hermeneutic nature emphasizes how participants make meaning of these experiences, with researchers engaging in a “double-hermeneutic” process by interpreting participants’ own interpretations. Third, its idiographic approach allows for careful attention to individual perspectives and experiences of mental health support [97]. This qualitative approach is particularly valuable for understanding the nuanced cultural and personal factors that influence how male PI adolescents perceive and engage with mental health supports.

#### 2.2.2. Integration of IPA with Talanoa

While IPA provides the methodological structure for understanding lived experiences, Talanoa offers a culturally appropriate way to facilitate these discussions within PI contexts. Talanoa’s emphasis on relational dialogue and collective meaning-making naturally complements IPA’s hermeneutic dimension. This blend of approaches honors both the analytical requirements of phenomenological research and the cultural values of PI knowledge-sharing practices.

#### 2.2.3. Participants

Following IPA methodology, participants were positioned as experts in their lived experiences [97]. Given that the study state’s population is only 0.90% PI (59), recruitment was conducted through PI family social media groups, a method shown to yield higher response rates and cost effectiveness [98].

The study included participants who identified as PI and male, between the ages of 13 and 18 years old, with English as their predominant language, and current enrollment in secondary school (either middle or high school). Nineteen male PI adolescents, aged 13–17, agreed to participate in the study. This sample size was determined using information power rather than saturation [99], considering the specificity of the study phenomena, the quality of participant dialogue, and the chosen analysis strategy. The diverse participant pool (detailed in Table 1) provided sufficient information power through the direct match between the study phenomena and participants’ lived experiences, making traditional saturation unnecessary [100,101]. This sample size also facilitated intimate group dynamics that balanced idea generation with meaningful engagement.

Prior to data collection, approval was obtained from the researchers’ university Institutional Review Board (IRB). Two weeks before the scheduled focus groups, parents received a comprehensive explanation of the study’s purpose, procedures, potential risks, and benefits, and were asked to review this information with their children. Prior to data collection, written informed consent was obtained from parents/legal guardians, and written assent was obtained from the adolescent participants. All participants were reminded of their right to withdraw at any time without penalty, and the key points were reviewed again immediately before the focus groups began. To protect participant privacy, confidentiality and anonymity were maintained using pseudonyms and secure data storage protocols. All documentation, including consent forms, assent forms, and IRB approval, are available upon request.

### 2.3. Data Collection

#### 2.3.1. IPA Focus Groups

While IPA traditionally employs individual interviews [97], focus groups have emerged as an effective data collection method for phenomenological studies [102,103,104] including PIs’ perceptions of mental health support [89]. Although focus groups may have limitations, such as the influence of peer dynamics and potential conformity to group norms [105], research suggests their effectiveness for sensitive topics. Participants are often more willing to share personal information in group settings than in individual interviews [106]. Focus groups foster group interaction, creating a comfortable atmosphere that encourages honesty and sincerity [107]. For vulnerable populations, focus groups provide a safe space to share ideas and opinions, addressing the shame and self-stigma often associated with mental health issues [90]. The smaller group size and assurances of confidentiality made participants more comfortable sharing their views on mental health support [102]. Given the limited research on PI male adolescents’ perspectives on mental health, we believed this approach would provide valuable insights into their views. Therefore, for the present study, three in-person focus groups were conducted to understand participants’ experiences and perceptions of mental health supports. This approach aligned naturally with the study’s Talanoa theoretical orientation, as both focus groups and Talanoa emphasize collective dialogue, shared meaning-making, and relational knowledge construction [91,92].

#### 2.3.2. Focus Group Procedures

A semi-structured, open-ended interview protocol was used during the focus groups. Prior to the study, the protocol was piloted with a group of PIs to refine and clarify the questions based on participant feedback [108]. This process led to adjustments in several questions, such as rephrasing for cultural relevance and simplifying language, to ensure clarity before the main study began.

All three focus groups followed an identical predetermined protocol that included both traditional interview questions and card sorting activities. Each focus group session used the same set of questions, such as “If someone who identifies as PI seeks out mental health services, how would they be viewed in their community?” To review the full interview protocols, see Appendix A. Within this standardized protocol, all three groups participated in two card sorting activities: one addressing mental health barriers and another exploring mental health services, with prompts derived from the existing literature. These card sorting activities were designed as hybrid sorts: participants were asked to rank cards featuring evidence-based statements about barriers to mental health services in order of significance [109,110,111,112]. Participants were also asked to categorize mental health services as either “helpful” or “not helpful”. This approach encouraged participants to engage with the material actively, fostering deeper dialogue and reflection. Research has shown that hybrid card sorts are particularly effective in eliciting communication among adolescents [113].

To encourage engagement, participants were given alternative response options, such as blank cards, ‘disagree’ cards, and ‘my opinion’ cards, allowing them to express unique perspectives. The primary goal of the card sorting exercise was to facilitate meaningful conversations about participants’ mental health experiences, aligning with the study’s focus on understanding their perceptions. The discussion focused on the factors participants deemed most significant as they categorized their cards, highlighting the qualitative and participatory nature of the method.

The flexibility of the card sorting process enabled participants to interpret and respond to the prompts in diverse and personal ways, enriching the data with a variety of perspectives. This approach also helped mitigate potential issues such as dominance by more outspoken group members, halo effects (where influential participants sway others), and groupthink (the tendency for individuals to suppress unique opinions to align with the group consensus) [102,114].

Each focus group was led by a facilitator accompanied by a notetaker to ensure thorough documentation and support during the sessions. The facilitator, a self-identified PI woman who practices as a school psychologist, brought valuable lived experience and professional expertise in mental health-seeking practices. The notetaker, who identifies as a Latino male (whom participants frequently mistook for Maori), contributed his experience as a special education teacher of adolescent males and current training in school psychology. Both team members completed over six hours of specialized training in interviewing techniques and focus group facilitation. The facilitator and notetaker began each session by sharing a meal with the participants, fostering rapport and setting a welcoming tone. At the start of the focus group, both the facilitator and notetaker warmly introduced themselves to create a comfortable and engaging environment. The facilitator adopted an informal, approachable style that encouraged open interaction among participants while skillfully redirecting conversations when necessary to ensure that everyone’s voice was heard. By using strategic follow-up questions and prompts for elaboration, the facilitator further built trust and created a safe space for participants to express themselves freely.

Each focus group lasted approximately an hour and a half, with a short break midway through. During the break, the facilitator and notetaker collaborated to ensure that all key points were recorded accurately. They also reflected on group dynamics and made careful observations about participants in each group as part of the data collection process. At the conclusion of the focus groups, participants were provided with information about mental health resources for themselves or their friends and family members. Notably, the term “mental health” was intentionally left undefined during the focus groups, allowing participants to express their perceptions and beliefs based on their own understanding of the term.

Focus group formation was guided by two primary factors: participant availability and the intentional creation of ethnically diverse groups. Rather than random assignment, we ensured each focus group included participants from different PI ethnic backgrounds (e.g., Samoan, Tongan, Hawaiian, Fijian, Maori) to capture a range of cultural perspectives within each discussion while working within the practical constraints of participants’ schedules.

### 2.4. Data Analysis

Focus group recordings were transcribed verbatim, including detailed notes on non-verbal communications and body language. Following transcript verification for accuracy, data analysis proceeded according to standard IPA procedures [97]. No formal hypotheses were formed during data collection to allow for unbiased examination of participants’ lived experiences [115].

The analysis followed these sequential steps:The transcripts were read and reread.Exploratory notes related to the research questions were created.Personal experiential statements from exploratory notes were developed (see Table 2).Exploratory notes were clustered into personal experiential themees for each focus group (see Table 3 and Table 4 for sample cluster tables for FG1).Personal experiential themes (PET) tables were created for each focus group.Personal experiential themeses across participants and focus groups were compared (see Table 5).A group experiential themes (GETs) table organizing categories from each focus group into four overarching themes, with subordinate themes listed as categories for clarity, was developed.

Finally, a synoptic quote table (Table 6) was developed, highlighting the four overarching themes.

The research team also crafted a narrative that captured both converging and diverging themes, maintaining essential qualities for rigor in using IPA methodology [116]. The analysis team included the focus group facilitator (PI female school psychologist), the notetaker (Latino male school psychology student with special education background), and a White female PhD researcher experienced in PI mental health research and teaching diverse adolescents in secondary public schools across multiple states in the USA.

### 2.5. Trustworthiness and Rigor

This study implemented multiple strategies to ensure methodological rigor and trustworthiness in data collection and interpretation [117], while adhering to established IPA quality markers [116]. The research followed Lincoln and Guba’s [118] standards of dependability, credibility, transferability, and confirmability through several key processes.

#### 2.5.1. Research Team Composition and Positionality

The core team included six researchers (4 female, 2 male) with expertise in qualitative methods, adolescent behavioral and mental health, and PI populations. Team members held various degrees (3 PhD, 1 EdS, 1 EdS Candidate, 1 MS-BCBA), with four members identifying as BIPOC (Samoan, Hawaiian, Tongan, and Latino/a/x) and two as White. Three members were practicing mental health providers. The team’s shared commitment to amplifying marginalized voices and personal connections to mental health challenges within PI communities informed their approach to the research.

#### 2.5.2. Trustworthiness Strategies

Trustworthiness strategies included expert review by six diverse reviewers (4 BIPOC, 2 White; 4 male, 2 female), including three PI PhD-level psychotherapists (2 Hawaiian, 1 Māori), who confirmed the findings’ plausibility [119]. Additionally, aligning with Talanoa traditions and our theoretical framework, member checking involved a community debriefing session. This session included participants; their families; and cultural, faith, and community leaders. During the session, a significant idea that was viewed as impacting PI mental health emerged: the concept of cultural homelessness. Participants, faith and community members, described this as the unique challenge PI adolescents face as they navigate between their families’ island traditions and the expectations of US public school culture. This collective dialogue and meaning-making process not only found our results plausible but also deepened our understanding of participants’ lived experiences. The research team also employed investigator triangulation to reduce potential biases [120] and conducted regular peer debriefing sessions to explore multiple interpretations of the data. An audit trail was maintained throughout the study, including training records and process documentation. The team’s diverse perspectives, both within and outside the PI community, also supported the interpretation of the data.

## 3. Results

The findings from this study, analyzed using Interpretative Phenomenological Analysis (IPA), revealed four overarching themes, each highlighting distinct yet interconnected aspects of PI adolescents’ mental health experiences. These themes provide insight into the cultural, relational, and systemic influences shaping their perceptions and approaches to mental health. Subordinate subthemes, organized as categories for clarity, further expand on the complexities within each area. Please note, typically, we would attribute all quotes to individuals. However, to maintain confidentiality and protect the anonymity of minors within this small community, we will reference quotes by focus group rather than by individual. Specifically, quotes will be annotated as FG1, FG2, or FG3 to indicate whether the speaker participated in focus group 1, 2, or 3, respectively.

### 3.1. Theme One: Stigma and Judgment

The theme of stigma and judgment emerged consistently across the three focus groups, suggesting cultural barriers to discussing and addressing mental health issues among PI Male Adolescents (PIMAs).

#### 3.1.1. Convergence and Divergence Across Focus Groups

All three focus groups emphasized the significant stigma associated with mental health challenges, which often discouraged open discussions and help-seeking behaviors. Participants commonly described mental health struggles as personal failings that brought shame not only to individuals but also to their families. They shared that mental health struggles were viewed as signs of weakness or incompetence and noted that cultural norms and familial expectations reinforced this belief, with statements such as “Polynesians don’t take mental health seriously” (FG2). The expectation to “suck it up” appeared consistently, with FG1 noting “for Polynesians sometimes when we struggle with your mental health like they don’t take it serious so like just tell them to suck it up” and FG2 echoing “sometimes my dad would be like, suck it up”. And, while all groups recognized stigma and judgment as pervasive barriers, “when they see that we do, like were seeking for help they’re just thinking like another good poly gone bad” (FG1), the way they contextualized it differed, with varying emphases and perspectives.

#### 3.1.2. External Consequences of Stigma and Community Standing

Focus Group 1 participants articulated deeply held concerns about how seeking mental health support could lead to permanent social consequences, affecting both individual and family standing within the PI community. Participants described immediate social consequences of seeking help, noting that “they’ll [someone with a mental health challenge] will probably not really talk to you as much cause they don’t feel like good about themselves, so they don’t really wanna talk to anybody”. The fear of social isolation was particularly evident in their description of friend group dynamics: “Once they know that you have to go to like a mental health place you’re being put off as like a bad person and that like no one will really want to hang out with you, your friend group will change”. While some friends might offer support, participants noted that “most people they’ll just like give up on you—they’ll leave”, leading to a painful process of discovering “who’s close to you who cares about you”.

Another unique aspect of FG1’s discussion was how mental health stigma intersected with athletic identity in the PI community. Participants explained, “the community that we come from like Polynesians are good at sports—all of us—we’re like good at sports”. This cultural expectation made seeking mental health support particularly risky, as it could mark someone as “another good poly gone bad”. The fear of disappointing these community expectations was palpable in their discussions, and the stigma was perceived by participants as long lasting: “for us [PIs] we’re [sports] players so recruiters will probably see us [someone with experiencing a mental health challenge] as they’re gonna to do more bad things as they come to college cause they have more access—so it will really affect us.”

Participants also emphasized how mental health challenges affected not just individual reputation but entire family standing. They described how seeking help could damage “the way you’re viewed and the way your family is viewed”, suggesting that mental health struggles were seen as reflecting poorly on family upbringing. The stigma could create a lasting family label, with participants noting that if you come from a family with mental health history, “you’ll just be known as another person who just like went down the wrong path and the wrong road”.

The group revealed how stigma influenced perceptions of personal responsibility for mental health. They described a cultural expectation that individuals should handle their own problems: “people can’t make decisions for you, it’s things that you do on your own”. Even when discussing help-seeking, participants emphasized personal responsibility: “if you do have a mental health problem then it’s not someone else’s job for them to look for you but if you need help you have to go explain it to them.”

Participants described how their community often minimized mental health challenges, noting “for Polynesians sometimes when we struggle with your mental health like they don’t take it serious so they like just tell them to suck it up”. They provided concrete examples of this dismissal, recounting how “people were making fun of them and people weren’t taking them really seriously”.

This group’s discussion of stigma particularly spotlighted how seeking mental health support was perceived as a marker of personal failure that could lead to permanent social consequences. Their concerns extended beyond individual reputation to encompass family standing, athletic identity and opportunities, and community expectations, revealing how PI male adolescents navigate social relationships and cultural expectations when considering mental health support.

#### 3.1.3. Unavoidable Reality of Judgment

While all focus groups discussed both stigma and judgment surrounding mental health challenges, participants in FG2 approached stigma as an inherent cultural trait rather than just a consequence of mental health help-seeking. Their discussion revealed a perceived acceptance of judgment as an inherent part of PI identity, emerging through a series of direct statements that built upon each other. Participants noted, “Anyone would be judged if they went to see a therapist and it got out”, followed by the frank admission “not gonna lie, PIs are very judgmental” and the straightforward declarations “we’re judgmental people though” and “it’s just the way we are”. These statements were not presented as complaints but rather as straightforward observations about their cultural reality.

Focus Group 2 also explicitly connected judgment to physical discipline and consequences, with participants casually noting “Ya like we got hit a lot” and matter-of-factly adding that if they disclosed mental health challenges “then I would be hit with a trash can”. They identified specific cultural expectations that created barriers to seeking help, describing how “It goes back to the culture, if it’s [mental health challenges] something you’re experiencing, [you should be able to] just roll over it, and you’ll be good by tomorrow”. They also recounted how some authority figures, particularly parents, would dismiss mental health concerns, noting “sometimes my dad would be like, suck it up”.

The group described specific labels used to stigmatize those with mental health challenges, sharing how people would question “What’s wrong with them?” or declare that someone “didn’t choose the right” or “is not grateful”. The pervasiveness of judgment led to self-protective behaviors, evidenced by one participant’s admission, “Sometimes I don’t say anything because I feel like people are going to judge me”. Yet despite their clear articulation of judgment as a cultural norm, participants expressed willingness to help others who struggled with their mental health, saying they would “help them out”, “comfort them”, and “talk to them to make sure they are okay”. This willingness to help others while avoiding personal help-seeking mirrors patterns seen in Focus Group 3.

#### 3.1.4. Helper–Avoider Paradox

Participants in Focus Group 3 revealed a complex dynamic that helping others (a longstanding strong PI cultural value) served as both a communal value and a potential deflection from personal struggles. This group uniquely emphasized collective support through concrete actions, evidenced by statements like “I was gonna have him [a friend with mental health challenges] move in with us cuz we had a guest bedroom in our house”, and organizing group assistance, “all my poly friends over there, like I talked to them about it and then they’re like, oh, have him come to [activities], help him out”.

The emphasis on being a helper emerged consistently through their discussions, with statements such as “See how you can help them”, “Just being there for them and inviting them to activities”, “Letting them know that they have people behind their back”, and “Just to be a good friend, just sit there and stuff like, you know, just listen”. However, this helper orientation appeared to function as a shield against acknowledging personal needs. This manifested most strikingly in statements that separated the self from those needing help: “there’s nothing wrong with me. I’m okay. And stuff. Like, why don’t I help someone else then?” This distancing occurred even while describing deeply personal experiences of support needed during difficult times, such as when one participant shared experiences with depression and suicidality: “I was really low, and thinking of doing some really stupid things”. And again, when another participant described the community response after his grandmother’s death: “they sent us like a Christmas song they were singing [the congregation] and they were just like there comforting us”.

Particularly notable was their framing of mental health challenges as something that someone could be “talked out of”, suggesting a view that mental health struggles were somehow optional or could be overcome through willpower: “Like talk them out of it. It’s up to them”. This perspective might help explain the strong preference for being a helper rather than help-seeker, as it positions mental health challenges as something one should be able to overcome independently. Interestingly, the group’s relationship with help-seeking is also reflected in how they maintained connections with those they assisted. The participants were quick to note that they remained friends with and continued talking to individuals they have helped with mental health struggles, as one shared, “I still talk to him now”. This dynamic emphasizes how adopting the role of “helper” provides young PI men with a culturally acceptable way to engage with mental health concerns. However, it also reinforces that the stigma around admitting their own need for support remains, making personal help-seeking more difficult.

#### 3.1.5. The Role of Humor in Navigating Mental Health Stigma

Humor emerged as a powerful yet complex cultural tool in managing mental health stigma across all three focus groups. Participants described humor as a shield against emotional pain, a way to maintain social bonds, and occasionally a barrier to addressing serious mental health concerns.

In Focus Group 2, participants openly discussed the dual nature of humor in their community. One participant acknowledged, “Ya, we make fun of each other because we think it’s funny, but some people might not think that”. This admission came alongside a broader acknowledgment of judgment within their culture: “Ya, I feel like we judge each other because we think it’s funny”. While humor was recognized as a way to strengthen social connections, it was also seen as a tool that could minimize or deflect serious concerns.

Focus Group 3 illustrated the real-time use of humor as both a coping strategy and a means of fostering comfort during emotionally heavy discussions. Even as some participants became emotional and cried when sharing personal struggles with mental health, light teasing and jokes helped ease the atmosphere. One participant explained, “You want to be hanging out with helpful people because they’d be funny. Just to keep your mind, like, you know, going. So, you don’t just sit there and be like, ‘Oh, shoot, man’”. Another elaborated on humor’s protective role, noting, “Like, you start thinking, like, you go through depression and stuff, and it makes you want to do stuff you don’t want to do. Like, you got a whole life ahead of you. It helps [friends who use humor and are funny] you get out of that”.

These discussions positioned humor as both a cultural strength and a potential obstacle. It allows PI male adolescents to connect with others and manage difficult emotions, but it can also deflect attention from underlying mental health challenges. This suggests that mental health interventions should embrace humor as a cultural strength while fostering opportunities for deeper, more serious engagement when needed.

### 3.2. Theme Two: Cultural Misalignment in Professional Mental Health Services

#### 3.2.1. Mistrust of Mental Health Therapists

Across focus groups, participants consistently emphasized the disconnect between their cultural experiences and the approaches of professional mental health providers, though the ways this mismatch was framed varied slightly. Therapists unfamiliar with PI values were perceived as lacking the cultural understanding necessary to address PIMAs’ struggles, particularly in contexts of family, community, and expectations.

Participants in FG1 expressed mistrust toward therapists, both out-of-culture and in-culture. They articulated that therapist often “don’t know the pressures you’re under or how culture plays a big role in who you are”. This sentiment was particularly evident in discussions about the high expectations placed on PI athletes. As one participant explained, “In [the islands], they expect you to go be a D1 [division one elite] athlete, and it just puts too much pressure on you”. These cultural intricacies, they argued, are critical to understanding their mental health challenges but are frequently overlooked or misunderstood by out-of-culture therapists. However, mistrust extended to in-culture therapists as well, with participants voicing concerns about sincerity and effectiveness. One participant noted, “Cause maybe like the person knows where you’re coming from, but he just like -what’s it called? Like say that you have a PI therapist and he knows what’s going on with you, but it doesn’t really matter to him and stuff like that”. This dual mistrust illustrates the complexities of seeking culturally relevant mental health support within their community.

Participants in FG2 expressed general mistrust toward therapists, particularly those unfamiliar with their cultural experiences. They stressed the importance of cultural understanding, with one participant emphasizing, “Knowledge, you don’t want to talk to someone who is dumb… you don’t want to someone who doesn’t know what they’re talking about or who has been through it”. They emphasized concerns about the therapists’ limited understanding of their distinctive pressures, stating, “Palagi [White] therapists don’t get it; they wouldn’t understand what we go through”. FG2 members also highlighted the contrast between how mental health struggles are handled in PI and non-PI communities. One participant noted, “The Palagi person, when they have mental health issues, everyone is going to comfort them. If a Polynesian kid had it, they’d wonder—what the crap!!” This sentiment extended to participant skepticism about the sincerity of out-of-culture therapists. As another participant in FG2 explained, “They’re [mental health providers] basically paid for what they’re doing, so I don’t know if they would really help me… these guys don’t care as deep like people who know more about us.”

Participants in FG3 expressed a layered mistrust of mental health therapists, indicating that therapy would only be considered as a last resort. Similar to FG2, they articulated that a lack of personal connection with therapists was a key barrier, stating, “Someone who doesn’t know me is the least helpful. Because I don’t want them telling everybody”. For these adolescents, trust had to be built over time, and therapists needed to demonstrate a genuine understanding of their personal and cultural backgrounds before they could be seen as a viable support option.

Therapy was also described as a fallback when other options, such as family or friends, had been exhausted. One participant explained, “There’s some ways that you can’t help [friend or family member with mental health challenges], and it’s probably better that you go to a professional and try and get help. Turn their ways to a professional therapist to help them. Just someone that can help you help them”. However, even in these instances, their confidence in therapists’ abilities to truly relate to their experiences remained low.

Like other focus groups, members of FG3 also discussed cultural misunderstandings that they believed therapists, even PI therapists, might perpetuate. For instance, one participant remarked, “If it has something to do with your parents too, they’ll be like, fight back or something like that”. Another added, “They would be like, This is your space. Just let him [parent] know. Dude! No! I can’t let him know”. These perceptions reflect the adolescents’ belief that therapists might provide advice that clashes with their cultural norms and family dynamics, such as suggesting confrontation or individualistic communication styles, which they felt were incompatible with PI values.

This sentiment of cultural disconnect extended to emotional expression shaped by life in the islands. Participants emphasized that their upbringing made it challenging to open up to professionals, as norms around emotional restraint and respect for authority differed from Western practices. While some participants acknowledged that a PI therapist might be more relatable, they expressed that even cultural insiders could fall short in connecting deeply. One participant noted that building a relationship with a therapist would take time, reinforcing how vital trust and familiarity are in the therapeutic process for PI male adolescents.

#### 3.2.2. Concerns About Confidentiality

Concerns about confidentiality emerged as another barrier to PIMAs seeking professional mental health support. Participants in FGs 1 and 2 expressed fears about their private information being mishandled or shared without their consent. In FG2, one participant likened the experience of confiding in a therapist to the children’s game “telephone”, stating, “You tell one thing to one person, and they just add a little bit more and tell everybody”. This fear of distortion and breach of confidentiality created apprehension about seeking help. Similarly, participants in FG1 highlighted the potential for rumors to spread after disclosing personal struggles in therapy. One participant explained, “If you do go to someone to get help, there could be rumors that start about you having mental health problems”. Another participant in FG1 reflected, “Some people think that when you go to like mental health places, your information can get leaked”.

#### 3.2.3. Perceptions of Financial Burden and Skepticism About Reliable Resources

The perceived financial burden of accessing professional mental health care emerged as a barrier across all focus groups, compounded by limited knowledge of available resources and skepticism about their reliability. Participants in FG1 described the economic strain of seeking mental health support, noting, “The cost and insurance is a barrier—because maybe you want to go, but your family doesn’t have enough money”. This sentiment was echoed in FG3, where participants discussed the sacrifices families would need to make, with one participant reflecting, “You feel like you’re putting your family deeper into a hole by seeking help”.

This perceived financial strain contrasted sharply with experiences in the islands. Participants explained that mental health support in the islands is often embedded within family or community networks and freely accessible. A participant in FG1 explained, “In [the islands], you don’t really go to mental health places; cost and stuff is all free because you go to family members, community members, so they’ll help you on their own time”. However, in mainland settings, the lack of such communal support systems left participants feeling unprepared and disconnected. One participant noted, “When I moved down here, [to the States], I felt like it was different… On [specific Pacific Island], they’ll definitely help you… But over here, it’s like all separated”.

In addition to financial barriers, participants expressed confusion about how to access care and where to seek help. For many, mental health services outside of religious communities and faith leaders (Is there anything else for support outside of the church and the bishop? [FG3]) were unfamiliar and intimidating. As one participant shared, “Polynesians like our culture, we don’t really go to mental health places, so we don’t really know where to go or where to find it”. [FG2] This lack of knowledge was further complicated by mistrust of the broader mental health system, particularly regarding medications and treatment options. Participants voiced concerns about unfamiliar medications, with one stating, “You don’t know what it is, so you don’t know if it will help you or put your life more at risk” [FG2].

The cultural disconnection from traditional island-based support systems and the unfamiliarity of navigating US mental health services created a sense of being caught between two worlds. Participants felt that while mental health services in the US were more formalized and resource-intensive, they lacked the cultural sensitivity and accessibility they were accustomed to on the islands.

### 3.3. Theme Three: Impact of Disclosure and Perceptions of Punitive Consequences

This theme captured the perceived risks and negative consequences PI male adolescents associated with sharing mental health challenges. Across focus groups, participants consistently expressed fears of stereotyping and punitive reactions, which deterred disclosure and reinforced the perception that vulnerability carries significant risks.

#### 3.3.1. Risks of Being Stereotyped

Participants described how disclosing mental health struggles could lead to being stereotyped as weak or a failure. In competitive environments, such as sports, this fear was heightened. One participant in FG1 explained, “If you’re an athlete, recruiters might see you as someone more likely to do bad things if they know you have mental health issues”. Similarly, another noted, “It makes it harder to accomplish your dreams because it holds you back from reaching your full potential” [FG1]. Cultural expectations compounded these fears, with participants articulating how their struggles reflected on their families. FG2 highlighted this dynamic: “When we fail, it reflects on our whole family. It’s not just about us—it’s about the name we carry”.

#### 3.3.2. Punitive Reactions to Vulnerability

Participants frequently described how disclosing mental health struggles often led to punitive responses rather than support or understanding. In FG3, participants shared that admitting to struggles could result in being “sent away to family members, hospitals, or rehabilitation centers to ‘figure it out’”. These punitive responses were sometimes framed as cultural discipline, intended to build resilience. One FG2 participant reflected, “You get beat up, and they’re like, okay talk to me tell me. Then it is Oh, I love you, beat you up. It’s really hard”. Similarly, another shared, “Growing up in [the islands], parents would kind of perceive you as, ‘Don’t tell me about your problems’”. Another participant said, “I’m kind of scared to share my feelings with my parents, cuz I, I’m gonna get beat if I share something about me or something that happened”. These practices left many adolescents feeling isolated and unsupported. One participant summed up the sentiment, explaining, “My parents would probably think about it [mental health support] as not being needed or unnecessary” (FG2).

This punitive dynamic extended to authority figures such as coaches, whose responses to mental health challenges varied. While some participants described coaches as trusted figures, willing to provide support during crises, others highlighted instances where coaches responded with discipline rather than empathy. In FG3, a participant humorously recalled being told to “get on the line” or to do push-ups, even at home, as a response to personal struggles. Another reflected, “My coach is helpful because he disciplines us during practice… but he’s not someone I would talk to because he yells a lot”. The duality of these relationships reflects the broader tension adolescents face when seeking help from authority figures—support can coexist with discipline, but the latter often prevents open dialogue about mental health.

#### 3.3.3. Emotional Toll of Dismissive Responses

Participants across all groups underscored the emotional toll caused by dismissive responses to their disclosures. These experiences often discouraged further help-seeking. In FG3, one participant explained, “If you pick your most trusted person and they blow it off, you just think to yourself, ‘No one else can help me,’ and bottle it up”. Similarly, in FG1, another participant shared, “It would really hurt because the first person you trust… if they don’t help, you feel like you just have to keep it to yourself”. This sentiment was echoed by another participant, who noted, “If they just blew it off, I would probably sink even deeper, but just keep to myself more, and it would be harder to tell the next person what’s really going on” [FG1].

This pattern of dismissive reactions reinforced a cycle of silence and self-reliance. Adolescents feared that sharing their struggles would lead to rejection, judgment, or punitive consequences rather than receiving meaningful support. One participant captured this tension, noting, “We have a stereotype that Polynesians are athletic, strong, and good at singing. If you do have mental health issues, people see you as not fitting the stereotype, like you’re worse than all the others” [FG1]. These stereotypes created additional barriers to disclosure, further isolating adolescents and deepening their struggles.

#### 3.3.4. Diverging Cultural Perspectives on Mental Health

Participants also highlighted the cultural barriers they faced when expressing emotions or seeking support. In FG2, one participant explained, “Growing up in the islands, telling your parents your true feelings is just not like… here in America”. This cultural disconnect posed significant challenges for PI adolescents, who often found themselves navigating between traditional values and the expectations of mainland mental health norms.

Many participants expressed feelings of being caught between these two systems, a sentiment akin to cultural homelessness. They faced pressure to uphold traditional values that prioritized resilience and self-reliance while simultaneously encountering Western mental health approaches that emphasized emotional expression and vulnerability. This duality often left them feeling unsupported in both contexts. As one participant in FG1 humorously summarized, when sharing mental health challenges, “Your family will use these techniques, like send you off to your aunties or uncles and your grandparents, in a different place where like they know you have nothings else better to do, (laughs) so yeah”. The consequence of cultural homelessness, compounded by the fear of punitive reactions, creates obstacles for PI adolescents in accessing meaningful mental health support.

### 3.4. Theme Four: A Complex Ecosystem of Trusted Relationships as Mental Health Support

Ultimately, PI male adolescents described a complex ecosystem of support that combined friends, family, faith leaders, and community members, despite systemic barriers and fears of disclosure. While each group played a distinct role, their collective involvement highlighted the essential value of trusted relationships in navigating mental health challenges and supports. FG3 highlighted the protective role of “trusted circles”, defining these as individuals who check in daily and maintain consistent emotional support. One participant shared, “People who text you, send you [funny things] they show they care”. This theme of consistent engagement as a trust marker was echoed in FG2, where participants described religious leaders as trustworthy because “as long as you aren’t planning to hurt yourself or anyone else—they don’t tell anyone what you share”.

#### 3.4.1. Friends as Initial Confidants

Participants across all groups noted that friends were often the first point of contact for discussing mental health concerns. One participant in FG3 shared, “Most friends will easily come up to you first instead of the parents ‘cause they feel more comfortable around you”. Friends were valued for their relatability and shared experiences, particularly among those with similar interests. Another participant explained, “If you’ve been friends with someone for a while and they don’t want to help, then you know you’ve been friends with the wrong people… friends should definitely help you with your problem”. Adolescents frequently expressed a preference for same-gender confidants, citing a sense of shared understanding and familiarity with their unique challenges [FG1, FG2]. One participant reflected, “I feel like if I were to talk to one of my guy friends, they would give more manly advice,’ like stick it out or tough it up, whatever”. Despite this, a few participants suggested female confidants might offer better support, as they perceived women to be more insightful, “Cause women know emotion more and like they’re more sensitive and so it could go both ways” [FG1, FG3]. Another participant agreed, “I feel like females are smarter and have more knowledge than men” [FG2].

#### 3.4.2. Complex Dynamics with Parents

Though PIMAs often said they would approach friends first when dealing with mental health struggles, as friends were where they felt the safest, parents were still recognized as critical sources of support. One major hesitation in approaching parents was the fear of being blamed, misunderstood, or dismissed. A participant explained, “You don’t want to open up to them… you feel like you messed up or did them wrong” [FG3]. Others expressed concerns about being told to “straighten up” or “man up”. As one participant described, “We all know what our parents would say. Just be like, oh, man up. Don’t be like that. You know—just don’t do that stuff and move on” [FG3]. This perception of parental responses created barriers to open communication, as adolescents felt their struggles might not be taken seriously.

Some participants also worried about being “babied” if their parents found out about their mental health concerns. “We’d be treated different… we’d get babied if one of our parents found out” [FG2]. Despite these challenges, there was a growing recognition that parents were beginning to adapt and learn more supportive behaviors. “Like now they’re starting to learn how to hear. They’ll try to support you and stuff with it because they are your family members” [FG3]. Another participant added, “Yeah, I feel like they would understand. But they would keep pushing you to your limits” [FG3], highlighting the complexity of parental support.

When participants reflected on the broader role of parents, many noted their deep value despite the initial reluctance to approach them. A participant shared, “At the end of the day, they still want to help you” [FG3], while another emphasized the value of parental experience: “I said parent because, maybe, you know, I don’t want to talk about some stuff with them, but still, they know you the best. So it could be really beneficial if you talk to your parent about stuff because they’ve been through a teenage life, they have experience, and that would help” [FG3].

Interestingly, when asked what they would do if a friend or relative come to them with a mental health issue, participants across all focus groups unanimously stated that they would go to the parents first. Some even mentioned involving faith leaders for additional support. This unanimous agreement seems to imply reliance on parents in times of crisis, even when adolescents expressed mixed feelings about approaching them for their own struggles.

#### 3.4.3. The Role of Faith Leaders

Faith leaders, particularly bishops, emerged as trusted sources for mental health guidance. Many participants emphasized the confidential and nonjudgmental support they received from bishops [FG1, FG2, FG3]. A participant in FG1 stated, “If someone came to me with a [mental health concern] I’d go to the bishop”. Another participant in FG2 explained, “They [bishop] always talk to you and don’t tell anybody what you’re going through unless it’s something really serious”. Still another participant in FG3 shared, “I went straight to the bishop” [when encountering a friend with a mental health concern]. Despite this trust, some adolescents expressed embarrassment about being seen speaking to a bishop, fearing others would assume they had done something wrong. As one participant noted, “If somebody saw you going into the bishop’s office, they’d probably think, ‘That person messed up real bad’” [FG2]. Even so, bishops were generally viewed as approachable and invaluable for spiritual and emotional guidance.

#### 3.4.4. Older Siblings and Grandparents

In addition to parents, other family members, particularly grandparents and older siblings, were highlighted as sources of support in mental health struggles. Participants frequently described the unique bond they shared with older relatives, particularly grandparents, who were often seen as wise and understanding. One participant shared, “My grandma was my go-to person to talk to; she understood me more than anyone else” [FG3]. Similarly, older siblings were valued for their ability to provide advice that felt more relatable and practical. One participant explained, “I tell a lot of things to my older brother than I do to my parents just because he’ll probably respond in a way that makes more sense to me than if my parents would tell me something because he just gives better advice in general” [FG3]. Another added, “Siblings could help too… they’ve been doing everything in this generation, so they could help more than your parents, cause your parents are older” [FG2].

Despite the crucial role grandparents and siblings play, participants acknowledged that traditional family dynamics could sometimes complicate open discussions about mental health. FG3 participants reflected on the cultural norms they experienced growing up in the islands, noting, “You don’t really share your feelings [there], but here it’s different—we’re expected to be more open”. This generational tension often made it challenging to bridge the gap between traditional values and the evolving norms of emotional expression in mainland contexts.

However, many participants noticed a positive shift in generational attitudes, particularly in how families are beginning to approach mental health. They shared optimism about the progression of mental health views within their families and expressed a desire to better support their own children in the future. One participant reflected, “But like now, since new generations, people just talk about each other’s feelings and stuff to parents and stuff. Not [PIs] yet. But like me, when I grow up and if my kid goes to depression and stuff, I’ll sit there and talk to them. Sometimes you gotta change other ways to change people, you know” [FG3].

#### 3.4.5. Broader Community Support

The importance of broader community networks was recurring across focus groups. All focus groups discussed the roles of coaches in the community, as was discussed earlier. Although some did not value coaches for mental health support, others did: “when you build up your relationship with your coach, they want to see you strive and make it to the next level and see you do good” [FG1]. Participants highlighted how community members often stepped in during crises. Relative to coaches, one participant shared, “He helped me when things were really hard at home and let me stay over at his house”. Other participants recounted the saving grace of community support during difficult times: “Then they found me on the street… and told me everyone was looking for me. It just showed me how much love and care everyone had for me” [FG3]. Community members, particularly those from sports and religious settings, provided a sense of belonging and security during difficult times. Another participant shared, “Being around the ward and finding good people is the greatest” [FG3].

## 4. Discussion

The results of this study can extend our understanding of mental health stigma in PI communities beyond generational differences previously identified in the literature [23,90,121]. While prior research has suggested that mental health stigma primarily resides with older generations in PI communities [89], our findings reveal that these attitudes are deeply internalized, even among adolescents, though they manifest in distinct ways.

Our findings also reveal a critical disconnect between adolescents’ expectations of adult support and the actual preparedness of adults in PI communities to address mental health concerns. While participants across focus groups indicated they would eventually turn to parents, coaches, or faith leaders for mental health support—believing these adults would “know what to do”—previous research suggests these adults often feel ill-equipped to provide such guidance [89]. This misalignment between youth expectations and adult capabilities creates a potentially dangerous gap in mental health support and is particularly concerning given the cultural preference of minoritized populations for seeking help within the community rather than from mental health professionals [85,122]. When adults lack mental health literacy but are positioned as primary support resources, there is a risk of perpetuating misinformation or inadvertently reinforcing harmful coping strategies. The situation is further complicated by the severe underrepresentation of PIs in mental health professions, with over 80% of licensed mental health professionals identifying as part of the ethnic majority group [123,124].

This also creates a troubling cycle: PI adolescents prefer to seek help from adults in their community, who may lack mental health knowledge, while simultaneously distrusting mental health professionals who lack cultural understanding. Breaking this cycle requires a two-pronged approach: increasing mental health literacy among trusted community adults while also addressing the severe lack of diversity in mental health professions [125]. Without such intervention, PI youth risk falling into a support gap where neither informal nor formal mental health resources adequately meet their needs. Our recruitment experiences for this study provided compelling evidence of this dynamic, with some families declining participation based on beliefs that mental health concerns should remain strictly within family boundaries. Such attitudes reflect deeper cultural values about family privacy and self-reliance but may also indicate internalized stigma about mental health discussion with outsiders.

The tension between community support and professional help-seeking creates a particular challenge for mental health interventions in PI communities. While the strong community bonds represent a valuable resource for mental health support, they may simultaneously reinforce isolation from professional services. This suggests that effective mental health interventions must find ways to build upon existing community strengths while creating bridges to professional services, rather than attempting to replace traditional support systems entirely.

The centrality of religious leaders in PI mental health help-seeking emerged as another finding in our study, extending beyond active religious participation. Notably, even participants who reported limited religious engagement identified faith leaders as trusted mental health resources, with some unaware of mental health support options beyond “church and the bishop”. This finding builds upon previous research documenting the high rates of religious participation among PIs [10], with approximately 90% reporting active faith engagement [126].

The universal recognition of religious leaders as mental health resources, regardless of personal religious involvement, suggests that faith leaders occupy a unique cultural position that transcends their strictly religious role. Their influence appears to stem not just from religious authority, but from their status as culturally aligned community leaders who understand PI family dynamics and cultural values [127]. This positioning makes them vital gatekeepers for mental health support, often serving as the first point of contact for families facing mental health challenges.

The widespread trust in religious leaders presents both opportunities and challenges for mental health intervention in PI communities [110]. While faith leaders’ cultural authority could be leveraged to reduce mental health stigma and facilitate professional help-seeking, their significant influence also means their personal attitudes toward mental health could substantially impact community perceptions [17]. This suggests that mental health outreach efforts should prioritize engaging and educating religious leaders, as their perspectives may shape help-seeking behaviors even among less religious community members.

Lastly, our findings revealed how traditional PI masculinity norms, particularly the warrior identity [89], intersect with cultural values around giving versus receiving help to create barriers to mental health support. The helper–avoider paradox emerged clearly across focus groups, where participants readily discussed helping others while expressing deep reluctance to seek help themselves [128]. This dynamic appears particularly pronounced for male adolescents due to cultural expectations of strength and self-sufficiency that align with traditional warrior ideals [78].

The financial aspects of mental health support add another layer of complexity to this dynamic. In PI communities, where status is often measured by one’s ability to give rather than accumulate resources, seeking mental health support presents a double barrier: it requires both accepting help (perceived as weakness) and utilizing family resources for personal benefit (perceived as selfish). Participants across all focus groups demonstrated acute awareness of the financial burden mental health services could place on their families, suggesting that help-seeking decisions are filtered through both masculine identity concerns and cultural values around resource allocation [88].

The interplay among masculinity, cultural values, and help-seeking creates particular challenges for male adolescents [129]. Our recruitment experiences highlighted this gender disparity, with one mother reporting that while her daughter attended therapy regularly, her sons refused to discuss mental health entirely. The ambivalence participants expressed about therapist gender preferences further illustrates this complexity. While some felt more comfortable discussing emotional topics with female professionals, others preferred male therapists who might better understand cultural pressures on PI men. This suggests that gender dynamics in mental health support extend beyond simple preferences to encompass deeper questions about cultural identity and expectations.

These findings suggest that effective mental health interventions for PI male adolescents must address not only practical barriers but also fundamental questions about how seeking help aligns with cultural ideals of masculinity and family responsibility [129]. Reframing mental health support to better fulfill family and community obligations, rather than as a personal indulgence, might help bridge this gap. Furthermore, understanding the nuanced role of gender in both help-seeking and service provision could inform more culturally responsive mental health services for this population.

### 4.1. Talanoa as a Methodological Framework

The use of Talanoa as a methodological framework proved particularly appropriate for this study, as evidenced by how participants naturally engaged in storytelling and relational dialogue when discussing mental health support. Across all three focus groups, participants demonstrated the cultural significance of narrative sharing, both as a means of providing support—“Everybody needs someone in their hard times… just sit there and listen” (FG3)—and seeking help within community contexts.

The framework’s effectiveness was particularly evident in how participants described mental health support occurring within collective settings. For example, when discussing helpful resources, participants often referenced communal spaces where stories could be shared safely, such as church gatherings or family meetings. As one participant noted, “Being around the ward and finding good people… it’s the greatest” (FG3). This aligns with Talanoa’s emphasis on collective dialogue and community-based problem-solving.

However, participants also revealed barriers to traditional Talanoa practices, particularly in intergenerational contexts. While younger participants expressed a desire for more open dialogue about mental health, they noted resistance from older generations, who often responded with dismissive phrases like “Man up, grow up, suck it up” (FG1). These findings suggest that while Talanoa remains a culturally relevant framework for mental health support among PI male adolescents, its effectiveness may be limited by evolving cultural tensions and generational differences in approaching mental health discussions. Our findings demonstrate both the value of Talanoa as a methodological framework and its potential limitations in addressing mental health concerns across generational boundaries in PI communities. These insights, combined with our broader findings about cultural disconnects and support systems, point toward several important implications for practice, while also suggesting areas where additional research is needed to better serve PI Male Adolescents seeking mental health support.

### 4.2. Implications for Practice

Our findings suggest several pathways for improving mental health support for PI Male Adolescents (PIMAs), with implications for multiple stakeholders in community mental health. Given the central role of religious leaders identified in our study, mental health interventions should prioritize engaging and training these trusted community figures. Religious leaders need support in developing skills to discuss mental health with youth while maintaining cultural sensitivity. Their unique position as cultural bridges makes them ideal partners in connecting community members with professional mental health services.

At the community level, interventions should focus on reframing mental health support in culturally appropriate ways. Rather than approaching mental health as an individual concern, programs should emphasize how mental wellness strengthens family and community bonds. Community awareness campaigns should address stigma while acknowledging cultural values around family privacy and self-reliance. These efforts should be developed in partnership with PI community organizations to ensure cultural authenticity and community buy-in.

Incorporating Talanoa into mental health interventions offers potential benefits for cultural relevance by emphasizing storytelling, relationality, and mutual understanding. When thoughtfully implemented, Talanoa can serve as a valuable tool for building trust and fostering open discussions about mental health. However, mental health practitioners must approach this cultural practice with respect and understanding rather than superficial application.

Our findings revealed important considerations for implementing Talanoa-informed approaches. While Talanoa-style dialogue can create comfortable spaces for mental health discussions, particularly among peer groups, practitioners must carefully navigate cultural dynamics. For example, PIMA participants expressed concerns about punitive consequences for disclosing mental health issues, especially in intergenerational contexts. These intergenerational tensions can significantly complicate traditional Talanoa practices, requiring practitioners to balance cultural traditions with contemporary mental health needs.

To ensure cultural authenticity, mental health practitioners should develop Talanoa-based interventions in partnership with PI community members who understand these cultural practices. This might include consulting with cultural experts, engaging community elders, and regularly evaluating whether implementations remain true to PI cultural values. Practitioners must be particularly mindful not to appropriate or oversimplify Talanoa, as doing so could diminish its cultural significance and therapeutic potential.

Additionally, for mental health professionals working with PIMA clients, building trust and ensuring confidentiality emerged as paramount concerns. Practitioners should clearly explain the Health Insurance Portability and Accountability Act (HIPAA) protections and confidentiality limits while adopting family-centered approaches that align with cultural values. Understanding the complex intersections of masculinity, cultural expectations, and mental health in PI communities is essential for effective practice. Also, professionals should consider how gender dynamics might influence the therapeutic relationship and be prepared to discuss these dynamics openly with clients.

Educational settings also play a crucial role in supporting PIMA mental health. Teachers and coaches should focus on building trusted relationships before attempting to address mental health concerns. These adults can model appropriate ways to engage in Talanoa about mental health, fostering safe spaces for such discussions within school settings. However, it is crucial that this is done with careful consideration of cultural norms and values.

Finally, professional development for mental health practitioners should emphasize cultural competency specifically related to PI communities. This includes understanding family dynamics, cultural values around giving versus receiving help, and the unique pressures faced by male adolescents in these communities. Success in supporting PIMA mental health requires carefully balancing traditional cultural values with contemporary mental health needs, always maintaining focus on the family and community contexts that are central to PI culture.

### 4.3. Limitations and Future Research

This study’s design aligns well with IPA methodology, which seeks homogeneous samples to provide deep understanding of specific lived experiences [116]. Our focus on PI male adolescents from the same geographic area with similar cultural backgrounds strengthened the study’s informational power and supports naturalistic generalization rather than statistical generalization [99]. While this methodological choice was appropriate for understanding this specific subgroup’s experiences with mental health help-seeking, it necessarily limits our understanding to this demographic. The findings, while rich and detailed for this population, cannot speak to the experiences of other PI groups, including female adolescents, those from different geographic regions, or those with different levels of cultural integration. Additionally, while our participants provided valuable insights into male adolescent perspectives, we cannot assume these experiences reflect those of PI adolescents in areas with larger PI populations or more established cultural support systems. Future research should explore mental health help-seeking among other PI subgroups, including female adolescents, those in areas with larger PI communities, and those with varying levels of cultural integration. Given the significant role of religious leaders identified in our study, research examining their perspectives on mental health support and their preparedness to serve as mental health resources would also be valuable.

## 5. Conclusions

This study extends our understanding of mental health support barriers among PI Male Adolescents (PIMAs) in several key ways. While previous research suggested mental health stigma primarily resided with older generations, our findings reveal these attitudes are deeply internalized by adolescents, though manifesting through concerns about future opportunities and family reputation. A disconnect emerged between adolescents’ expectations of adult support and adults’ actual preparedness to provide mental health guidance, creating a troubling cycle where youth prefer to seek help from potentially ill-equipped community members while distrusting professional services. The central role of religious leaders, even among less religious participants, suggests their potential as key mental health gatekeepers. Additionally, traditional masculinity norms intersecting with cultural values around giving versus receiving help created unique barriers, particularly when combined with financial concerns about mental health support. The effectiveness of Talanoa as a methodological framework highlighted the importance of storytelling and relational dialogue in PI communities, though generational tensions may limit its current effectiveness in mental health discussions. These findings suggest that effective interventions must simultaneously address practical barriers while engaging with deeper questions about how seeking help aligns with cultural values and family responsibilities.

## Figures and Tables

**Table 1 ijerph-22-00062-t001:** Male Adolescent PI Participants.

Participant	Age	Ethnicity
Participant 1	16	Native Hawaiian/Samoan
Participant 2	14	Native Hawaiian/Samoan/Tongan
Participant 3	14	Native Hawaiian
Participant 4	15	Samoan/Tongan
Participant 5	14	Samoan
Participant 6	17	Samoan/Tongan
Participant 7	16	Samoan
Participant 8	15	Samoan
Participant 9	15	Samoan
Participant 10	14	Native Hawaiian
Participant 11	16	Samoan
Participant 12	17	Maori/Samoan
Participant 13	14	Maori
Participant 14	15	Native Hawaiian
Participant 15	13	Samoan
Participant 16	17	Samoan/Tongan
Participant 17	15	Native Hawaiian/Samoan
Participant 18	14	Tongan
Participant 19	14	Samoan

**Table 2 ijerph-22-00062-t002:** Experiential Statements.

IPA Step 5 Create Experiential Statements	
Research Question(s): 1. How Do Pacific Islander Male Adolescents Experience and Make Sense of Mental Health Support Within Their Cultural and Social Contexts?	
Exploratory notes	Experiential statements
If you have mental health challenges you get sent to a mental hospital, a rehab, or sent away by your family to get yourself together and figure out your problems.	Mental health challenges are viewed as personal failures, leading to feelings of embarrassment and shame.
I would feel so embarrassed because now people will think I’m an addict or someone who made really bad choices.	People who disclose mental health issues are often judged as weak or incapable.
MH issues are your own fault because you chose to make the decisions that you did.	Mental health struggles are often seen as the individual’s fault due to poor decisions.
PIMA feel MH is very punitive—you did something wrong, so if you have MH challenges, you need to straighten up and get back on the right path.	
If you have a MH challenge, you’ll be viewed as getting caught doing things you should not do.	

**Table 3 ijerph-22-00062-t003:** Sample Cluster Table 1 for FG1.

PET 1 Stigma and Judgment	PET 2 Distrust and Barriers in Professional Help	PET 3 Family Support	PET8 Punitive View of Mental Health
Exploratory note Disclosing mental health concerns can lead to stereotypes that damage one’s reputation or opportunities.	Exploratory note Therapists from outside the Polynesian culture are perceived as unable to understand their unique pressures and experiences.	Exploratory Note Families, especially close relatives or grandparents, are viewed as primary sources of support in times of mental health struggles.	Exploratory note Mental health challenges are equated to moral or personal failure, making adolescents hesitant to seek help.
Explanation This category reflects the cultural and social attitudes that perpetuate the stigma surrounding mental health challenges. Mental health issues are often perceived as personal shortcomings or failures, leading to embarrassment and shame. This stigma discourages open dialogue and prevents individuals from addressing their mental health needs effectively.	Explanation This category addresses the skepticism and barriers Pacific Islander adolescents face when considering professional mental health services. Concerns about confidentiality, such as the fear of information being leaked, discourage seeking help. Financial constraints and limited knowledge about accessing mental health resources also serve as significant barriers, preventing individuals from pursuing professional care.	Explanation This category highlights the influence of family relationships on mental health disclosure and support. The family plays a dual role as both a potential source of support and a barrier due to cultural norms, fear of judgment, or punitive responses.	Explanation This category highlights the perception of mental health challenges as moral or personal failings. Community members may teach their children to avoid those with mental health struggles, perpetuating exclusion. Authority figures often treat mental health as a problem to fix rather than an issue requiring empathy. This punitive view discourages help-seeking and fosters an environment of judgment and blame.

**Table 4 ijerph-22-00062-t004:** Sample Cluster Table 2 for FG1.

PET 4 Role of Authority Figures	PET 5 Coping Mechanisms and Solutions	PET 6 Generational and Cultural Tensions	PET 7 Impact of Disclosures and Punitive Responses
Exploratory note The church and spiritual leaders, like bishops, are trusted sources for guidance and support.	Exploratory note People often rely on faith, spiritual guidance, or familial care as primary coping mechanisms for mental health struggles.	Exploratory note There’s tension between traditional parenting styles and newer approaches to mental health support.	Exploratory note Coaches and role models may take a punitive approach, focusing on discipline rather than support.
Explanation This category discusses the influence of authority figures, such as coaches, teachers, and church leaders, on adolescents’ mental health. Authority figures play a significant role in shaping adolescents’ views and approaches to mental health but may unintentionally perpetuate stigma or barriers.	Explanation This category explores how Pacific Islander adolescents cope with mental health struggles. While they encourage others to share their challenges, many avoid disclosure. Help-seeking is often a last resort. Faith and familial care are primary coping mechanisms, reflecting a reliance on internal and community-based solutions.	Explanation This category examines the impact of generational and cultural shifts on mental health perceptions. Younger generations feel a disconnect from traditional expectations of stoicism and discipline. Polynesians in the US face unique challenges adapting to these norms, often feeling alienated from both their cultural roots and the dominant culture.	Explanation This category captures the negative consequences of sharing mental health struggles. Disclosing challenges can lead to stereotyping, discrimination, and missed opportunities. Adolescents fear losing trust or being viewed as a liability. When disclosures are mishandled, it can deter future help-seeking behavior. This reinforces the perception that vulnerability carries significant risks.

**Table 5 ijerph-22-00062-t005:** Group Experiential Themes (GETs).

GET 1: Cultural Stigma and Judgment Theme Summary: Stigma Around Mental Health Is Deeply Rooted in Cultural Norms, Humor, and Societal Expectations Within the Pacific Islander Male Adolescent (PIMA) Community. This Stigma Often Prevents Open Discussions About Mental Health, as Such Struggles Are Seen as Personal Failures or Weaknesses. PIMA Can Help Others as Culturally Appropriate and Expected But Do Not Ask for Help. Seems to Be Cultural Expectation.	
Category	Quotes
1a. Mental health challenges are perceived as personal failings, leading to shame and embarrassment.	Everyone knows that when you need mental help… you’re just too embarrassed. You don’t want to make a bad reputation for yourself. [FG1]
	Anyone would be judged if they went to see a therapist and it got out. They’d be judged [FG1]
	Not gonna lie, pacific islanders are very judgmental. [FG2]
1b. Disclosing struggles often results in gossip, ostracism, or being labeled as weak or incapable.	The community that we come from, like Polynesians are good at sports. When they see us seeking help, they just think, ‘another good Poly gone bad’—and then the gossip spreads. [FG1] People would say What’s wrong with them? [FG2] He didn’t choose the right or -anything like that. [FG2]
1e. Cultural pride adds a barrier, help others—avoid being helped for MH challenge FG1, FG2, FG3	Like that good feeling like helping someone out like makes you feel good and stuff. Everybody needs someone in their like hard times and stuff, so, you know. Just to be a good friend, just sit there and stuff like, You know, just listen, probably focusing.

FG—focus groups.

**Table 6 ijerph-22-00062-t006:** Synoptic Quote Table.

Theme	Category	Quote
Stigma and judgment	External consequences of stigma and community standing	My community if you have a mental health issue, they view you as like someone less than you were.
	Unavoidable reality of judgment	They’ll just see us as one of those good stars that just threw their life away.
	Helper–avoider paradox	There’s nothing wrong with me. I’m okay. And stuff. Like, why don’t I help someone else then?
	The role of humor in navigating mental health stigma	You want to be hanging out with helpful people because they’d be funny. Just to keep your mind, like, you know, going. So, you don’t just sit there and be like, ‘Oh, shoot, man’.
Cultural misalignment in professional mental health services	Mistrust of mental health therapists	[Therapists] don’t help you with your culture like if they don’t know what your culture is.
	Concerns about confidentiality	If you go to like someone to get help there could be rumors that start about, he has mental health problems.
	Perceptions of financial burden and skepticism about reliable resources	Maybe you want to go but your family doesn’t have enough money…so you’re just putting your family deeper down.
		Our culture we don’t really go to mental health places, we don’t really know where to go or where to find it.
Impact of disclosure and perceptions of punitive consequences	Risks of being stereotyped	Once they know that you have to go to like a mental health place you’re being put off as like a bad person.
	Punitive reactions to vulnerability	You get beat up, and they’re like, okay talk to me tell me. Then it is ‘Oh, I love you, beat you up.’ It’s really hard.
	Emotional toll of dismissive responses	You pick your most trusted person, they don’t help you, so you just think to yourself, no one else can help me from here on out.
	Diverging cultural perspectives on mental health	Growing up in the islands, telling your parents your true feelings is just not like… here in America.
A complex ecosystem of trusted relationships as mental health support	Friends as initial confidants	What most polys would do is try to get like try to solve their problems with friends and stuff.
	Complex dynamics with parents	They’ve lived a teenage life; they know what we could be going through.
	The role of faith leaders	You don’t want to open up to them… you feel like you messed up or did them wrong.
	Older siblings and grandparents	He’ll just get you on the right path and he’ll always put God first and revelation from Him and church is a really big thing in my life.
	Broader community support	Close family members who know the struggle that you’re going through, they will help you.
		Siblings [older] could help too… they’ve been doing everything in this generation, so they could help more than your parents, cause your parents are older.
		When you build up your relationship with your coach, they want to see you strive and make it to the next level and see you do good.

## Data Availability

The datasets presented in this article are not readily available due to IRB restrictions protecting the confidentiality of vulnerable participants, including minors discussing sensitive mental health topics. Requests to access the datasets should be directed to Elizabeth A. Cutrer-Párraga at ecutrer@byu.edu, though access may be restricted based on IRB requirements and participant protection protocols.

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
