# Peer review of "“It Would Ruin My Life”: Pacific Islander Male Adolescents’ Perceptions of Mental Health Help-Seeking—An Interpretative Phenomenological Analysis Focus Group Study"

_ijerph, 2025, doi:10.3390/ijerph22010062_

Round 1

Reviewer 1 Report

Comments and Suggestions for Authors

Please review the use of abbreviations throughout the article. The standard practice is to spell out the term fully the first time it appears (e.g., "United States") and then use the abbreviation consistently thereafter (e.g., "US"). For instance, in lines 29, 36, 50, 64, and 150, the usage alternates inconsistently between "US" and "United States." Ensuring consistency will enhance clarity and professionalism.

In line 136, it would be more appropriate to use the full term "Attention Deficit Hyperactivity Disorder (ADHD)" when it is first introduced, following standard academic conventions. This will ensure clarity for readers who may not be familiar with the abbreviation.

In line 155, the phrase "the AAPI community, including PIs" is redundant, as Pacific Islanders (PIs) are already encompassed within the term "AAPI." Consider revising this to avoid redundancy and improve precision.

In line 192, the acronym "CDC" is mentioned, but its full form is not provided. Please ensure that all acronyms are spelled out the first time they appear in the text for clarity. For instance, if "CDC" refers to the Centers for Disease Control and Prevention, it should be specified.

Since the study is qualitative in nature, it is essential to clearly outline the study’s objective and any hypotheses at the end of the introduction. This will provide readers with a clear understanding of the study’s purpose and the questions it aims to address. Including these elements will also enhance the focus and coherence of the research.

In section 2.2, it is important to clearly specify the type of study being conducted. This will help readers understand the methodological approach and framework of the research. Clearly stating whether the study is qualitative, quantitative, or mixed-methods, for example, would provide more context and clarity.

In lines 305-306, it is mentioned that two focus group questions incorporated a card sorting activity based on literature-based prompts about mental health barriers and services. However, since three focus groups were used, it is unclear why the activity was applied differently in these two groups. Additionally, there is no distinction made in the results section regarding the variation in the focus group activities. Please clarify why this approach was taken and ensure that the results reflect any differences in methodology if applicable. Also, could you specify how the focus groups were formed? Were the groupings random, or was there a specific criterion for their selection?

In line 436, "FG2" is used, but the meaning of the abbreviation "Focus Group 2" is not indicated. It is important to spell out the term fully the first time an abbreviation is used in the text, especially since "focus group" is mentioned multiple times without abbreviation before this. Please ensure that all abbreviations are clearly defined when first introduced to avoid confusion for the readers.

Overall, this is a well-done and necessary study, addressing an important issue. However, I would have appreciated a comparison between genders to explore how females specifically face these circumstances, as mentioned in the future research of the article. 

I noticed that some references are missing their DOI. Please ensure that all references with available DOIs are included, as this will help improve the accuracy and accessibility of the citations. 

REF 3 -- DOI: 10.12968/bjsn.2015.10.1.19

REF 10 -- DOI: 10.1037/a0023266

REF 43 -- DOI: 10.1300/J035v20n04_03

REF 49 -- DOI: 10.1177/1077558709335173

REF 70 -- https://doi.org/10.1023/A:1016820106641

REF 92 -- doi: 10.1111/apv.12061

REF 117 -- DOI: 10.1177/001440290507100205

REF 128 -- DOI: 10.1037/cou0000185

REF 129 -- DOI: 10.1177/10105395211022944

Reviewer 2 Report

Comments and Suggestions for Authors

This is a well written paper with clear evidence of rigor within the design and analysis of the study. The introduction, methodological and discussion sections stand out as the strongest. There are however a few minor recommendations to further strengthen the paper. These include:

1. Merging of sections 1.4.3 and 1.4.3. This will be recommended for 1.5, 1.6 and 1.6.1. Once merged the sections can be made more succinct. 

2. More is needed on the culturally grounded PI framework. It is listed as a theoretical framework and then further discussed as a method in a subsequent section. This can be returned to within the discussion to speak to the implications for further research. 

3. The findings are quite extensive and can be strategically rewritten to choose 3-5 of the most salient points for presentation.  This would allow for further engagement of the conversation within the focus group interviews as well. 
